# Cons2Plan: Vector Floorplan Generation from Various Conditions via a Learning Framework based on Conditional Diffusion Models

## ABSTRACT

The field of floorplan generation has attracted significant interest from the community. Remarkably, recent progress in methods based on generative models has substantially promoted the development of floorplan generation. However, generating floorplans that satisfy various conditions remains a challenging task. This paper proposes a learning framework, named *Cons2Plan*, for automatically and high-quality generating vector floorplans from various conditions. The input conditions can be graphs, boundaries, or a combination of both. The conditional diffusion model is the core component of our *Cons2Plan*. The denoising network uses a conditional embedding module to incorporate the conditions as guidance during the reverse process. Additionally, *Cons2Plan* incorporates a two-stage approach that generates graph conditions based on boundaries. It utilizes three regression models for node prediction and a novel conditional edge generation diffusion model, named CEDM, for edge generation. We conduct qualitative evaluations, quantitative comparisons, and ablation studies to demonstrate that our method can produce higher-quality floorplans than those generated by state-of-the-art methods.

## CCS CONCEPTS

• **Applied computing** → **Computer-aided design**; • **Mathematics of computing** → Probabilistic reasoning.

## KEYWORDS

Vector Floorplan Generation, Graph Generation, Conditioned Diffusion Model

## 1 INTRODUCTION

Automated vector floorplan generation has experienced remarkable growth due to the surge of deep learning techniques [4, 24, 35]. These approaches can be divided into two categories. The first type, such as *RPLAN* [36], predicts floorplan images at the pixel level using boundary conditions. The second type, exemplified by *Graph2Plan* [15], generates floorplans at the box level by determining room-bounding boxes. This method uses boundaries or integrates graphs for conditional floorplan generation. Despite their effectiveness, these two methods require an additional post-processing step to obtain vector floorplans and the resulting floorplans lack variations. Recently, a box-level floorplan generation method named *HouseDiffusion* [29], which is based on a continue

*ACM MM, 2024, Melbourne, Australia*
© 2024 Copyright held by the owner/author(s). Publication rights licensed to ACM.
ACM ISBN 978-x-xxxx-xxxx-x/YY/MM
https://doi.org/10.1145/nnnnnnn.nnnnnnn

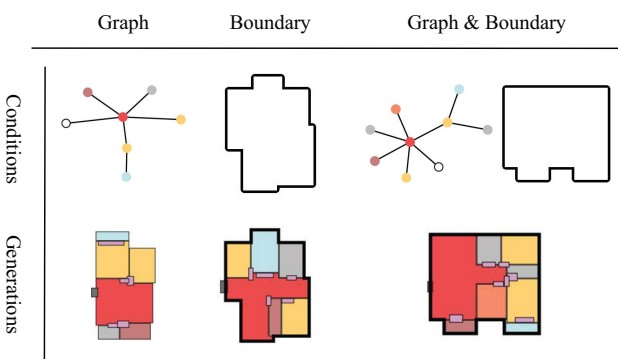

**Figure 1: Given three different conditions as input, Cons2Plan directly generates the vector floorplans without any post-processing. Users can generate floorplans by simply adjusting two parameters based on their desired conditions.**

and discrete diffusion model, has eliminated the need for any post-processing steps. Additionally, the probabilistic sampling of the diffusion model also contributes to the diversity of floorplans. However, *HouseDiffusion* is limited to using graph conditions and cannot generate floorplans under boundary conditions and their combination. In the actual design process, the boundary condition is frequently considered preliminary information for designers when crafting floorplans. Furthermore, boundary conditions help ensure that the floorplan efficiently utilizes the available space within a given area and can influence the overall appearance and proportion of the floorplan [15].

In this paper, we propose a learning framework named *Cons2Plan* for floorplan generation under three different conditions (see Figure 1). *Cons2Plan* is inspired by *HouseDiffusion* and builds upon its foundation by adding the latter two condition options. The key idea of *Cons2Plan* is employing a conditional diffusion model with a carefully designed conditional embedding module in the denoising network (see Figure 2). We transform the conditional diffusion model into a floorplan generator by augmenting its underlying transformer encoder backbone with cross-attention [33], which has been proven to be an effective method for handling various input modalities [16, 17]. In the input conditions, since our learning framework always requires the graph condition as a conditional input, generating the graph condition from the boundary becomes necessary when only the boundary is provided. Moreover, generating graphs based on the boundary is a useful yet overlooked task in architectural designs. So, *Cons2Plan* also incorporates a two-stage approach to generate graphs based on the boundary image. Our two-stage approach consists of two steps: (i) the first step contains three regression networks to predict the number and types of rooms in floorplans based on boundary conditions, and (ii) the second step includes an edge generation network conditioned on boundaries

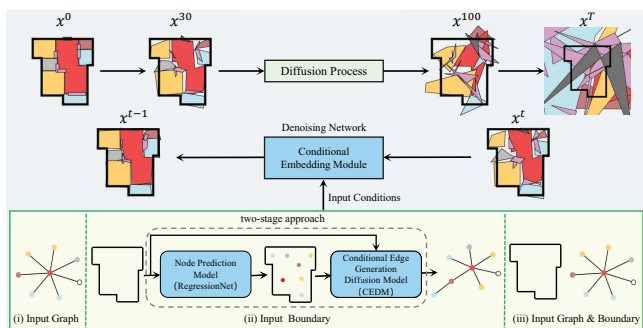

**Figure 2: The overview of our learning framework. It consists of two primary components: one part is the denoising network of the conditional diffusion model, and the other is a two-stage approach that uses boundary images as input to generate diverse graphs.**

and previously predicted room information, which is implemented using a conditional edge generation diffusion model, named CEDM. The approach leverages the regression model to predict room information while thoroughly considering boundary conditions and uses the diffusion model to ultimately generate diverse and high-quality graphs.

Extensive experiments demonstrate *Cons2Plan*'s ability to effectively generate floorplans based on different conditions and outperform the existing state-of-the-art methods (e.g., *RPLAN* [36] and *Graph2Plan* [15]) in both qualitative and quantitative evaluations. In scenarios with more constraint conditions than in *HouseDiffusion* [29], the performance metrics are nearly identical.

**Our Contributions.** The main contributions of our work can be summarized as follows.

- We propose a learning framework based on a conditional diffusion model that can automatically and efficiently generate vector floorplans considering three conditions.
- We design a two-stage approach that generates plausible and diverse graphs based on the input boundary conditions, serving as an important component of our learning framework.
- Extensive experiments demonstrate the superiority of our method under various input conditions. Furthermore, we conduct ablation studies to illustrate the effectiveness of the proposed conditional embedding module and two-stage approach.

## 2 RELATED WORK

### 2.1 Learning-based floorplan generation

Most of the existing floorplan generation methods are either pixel-level generation methods or box-level generation methods.

**Pixel-level floorplan generation**. [36] presents a two-stage strategy, named *RPLAN*, to automatically generate floorplans by first computing the number and types of rooms, and then determining the positions of the walls between the rooms. [25] propose a generative model named *House-GAN* for floorplan generation. *House-GAN* utilizes a generative adversarial network that takes graphs as input to generate diverse floorplans that fit the graph. Subsequently, they made improvements based on *House-GAN* and

proposed *House-GAN++* [26]. *House-GAN++* combines a relational GAN with a conditional GAN. Like other pixel-level methods, post-processing is still required afterward. Most importantly, the quality of the floorplans generated by it is significantly lower compared to *HouseDiffusion* [29].

**Box-level floorplan generation**. [15] achieves a learning framework called *Graph2Plan* that generates floorplans using graphs and boundaries as constraints. This framework heavily relies on the dataset, as it uses the Turing function to search for the most similar boundary within the dataset. Moreover, the post-processing step is also essential. [5] uses a generative adversarial network and graph convolutional network to produce floorplans from linguistic descriptions. [35] proposes a learning-based method that generates floorplans by combining suitable rooms together. Recently, [29] uses a generative model for floorplan generation through a diffusion model named *HouseDiffusion*. Taking a graph as input, *HouseDiffusion* can generate diverse vector floorplans that fit the graph and require no post-processing. However, there is still a drawback: the input conditions are singular, as they only accept graphs as constraint conditions.

### 2.2 Graph generation

Graph generation techniques have been extensively applied in various domains, such as social networks [7, 40] and chemical compounds [30]. Among these, state-of-the-art approaches use deep neural networks for graph generation. [39] devises a reversible mapping between the latent space and the graph, which generates node feature and edge feature matrices for the graph. [8] designs a GAN-based graph generative model in which the purpose of the discriminator is to ensure that the generated graph contains the required properties. [34] first uses discrete diffusion models, named *DiGress*, to generate graphs. *DiGress* adds noise to vertices and edges and predicts the types of vertices and edges. However, existing graph generation methods have not been directly applied to generate graphs in floorplans, nor have they generated plausible graphs with boundaries as constraint conditions.

### 2.3 Conditional diffusion models

Recently, the Diffusion Model (DM) [3, 9, 20, 37] has gained significant attention. Subsequently, to increase controllability, the Conditional Diffusion Model (CDM) has also been proposed. [10] introduces a Conditional Diffusion Model that integrates a classifier into the sampling process to guide the generation process. [22] further developed this approach by augmenting the number of control conditions. Later, [14] introduced an approach that directly embeds conditions into the denoising network to control the generation process. This method has a higher training cost but significantly better performance.

However, there are few works on using CDM for floorplan generation. Some works [2, 6, 21] use discrete 2D coordinates as the object. However, their methods are based on unconditional generation using DM. More recently, [29] first proposes a vector floorplan generation method based on the CDM, named *HouseDiffusion*. It can directly generate high-quality and diverse floorplans, even though the graph is the sole input condition provided.

# 3  METHODOLOGY

In this section, we first discuss the representation methods for floorplans and conditions in Section 3.1. Then, we introduce the two-stage approach to generate graphs, which serves as an essential part of our framework, in Section 3.2. Finally, we introduce the conditional embedding module in Section 3.3.

## 3.1  Floorplan and conditions representation

For floorplan data representation, we represent a floorplan as a set of axis-aligned polygonal geometries $G = \{G_1, ..., G_i, ..., G_N\}$. Where N denotes the total number of rooms and doors. Each polygonal geometry is defined as a sequence of corners with 2D coordination $G_i = \{C_{i,1}, C_{i,2}, ..., C_{i,N_i} | C_{i,j} \in R^2\}$. $N_i$ denotes the number of corners in $G_i$, which needs to be specified during sampling. Since the number of corners greatly influences the quality of the generated floorplan in satisfying the boundary conditions, we set the corner count for all room types, excluding living rooms, to 4. Upon examining the dataset, we observe that 95% of non-living room-type rooms have four corners, making this a reasonable generalization. The living room corner count, however, is sampled from the probability histogram during each inference. This histogram is constructed based on the dataset.

For graph condition representation, the graph is illustrated using a bubble diagram where nodes represent rooms, and edges indicate room connections. Each room node is associated with a specific room type. Furthermore, we use adjacency matrices for storing and processing the graph condition.

For boundary condition representation, we represent it as a three-channel image. The three-channel image includes the following information at each pixel, which defaults to 0:

- *Inside mask*: taking a value of 1 for the interiors.
- *Boundary mask*: taking a value of 0.5 for the exterior walls.
- *Global mask*: Combining inside and boundary masks together.

## 3.2  Two-Stage approach for graph generation

The graph generation based on the boundary plays a crucial role in both architectural design and our learning framework. Although there are many methods capable of implementing graph generation, none of these methods consider generating graphs conditional on boundaries. Therefore, we propose a graph generation method that fully considers boundary conditions. For the conditional graph generation method, inspired by [18], we formulate the graph generation problem as node prediction and edge generation. The process of generating graphs from a building boundary involves two stages: predicting nodes of the graph and obtaining edges between nodes. Node prediction is implemented by three regression models and edge generation is implemented by a conditional diffusion model called CEDM.

**Node prediction.** Our first step is to predict the number and types of rooms by using the building boundary. The node prediction proposed by [36] can accomplish this task quite well, so we directly use their three networks. We first use the LivingNet, a regression network, to determine the living room's location. We then adopt LocationNet, an encoder-decoder network, for predicting the type and position of the next room to be added. Upon obtaining the type of the next room to be added, we use ContNet, a regression network,

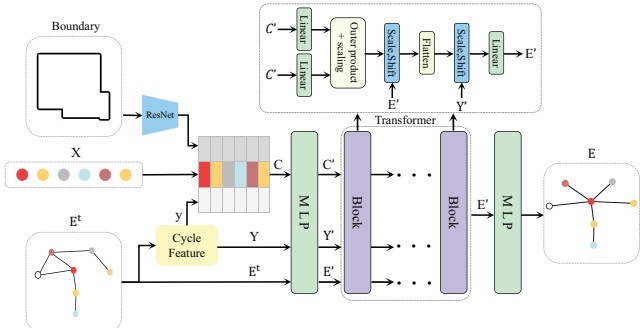

**Figure 3: The overview of CEDM. It tasks as input boundaries, node sequence $X$ and noisy edge $E^t$, and outputs clean edge. Our transformer blocks also feature residual connections and layer normalization. Scale and Shift operation is $(X_1 M_1 + 1) \odot X_2 + X_1 M_2$ for learnable weight matrices $M_1$ and $M_2$.**

to determine whether to continue adding rooms. We only use the predicted number and types of rooms, not their positional information, because restricting the locations of rooms would reduce the diversity of the generated graphs.

We modify the data input format for the networks by using a three-channel image as the input, which is consistent with the boundary condition representation. The architecture of the three networks has not been modified. The resulting node information will serve as the input for the next step.

**Edge generation.** The next step is to generate edges. There are some methods based on RNN [38] or GAN [8] that can implement edge generation. However, they have been proven to be less effective in terms of accuracy and diversity compared to methods using Diffusion Models [34]. Inspired by the graph generative model (Discrete Graph Denoising Diffusion model, *DiGress*) [34], we use a modified *DiGress* to generate edges. Unlike *DiGress*, which predicts the probability distribution of node attributes and edge attributes in the graph simultaneously, we use node attributes and boundary as conditions and focus on predicting the probability distribution of edge attributes. Therefore, our model is a conditional edge generation diffusion model named CEDM.

The reason we do not use *DiGress* to directly generate graphs with boundaries as conditions is that we discover *DiGress* cannot predict the number of nodes in a graph. Instead, the number of nodes is pre-sampled from a histogram of node counts derived from the dataset, without considering the boundary conditions. Existing DM-based graph generation models all use the same strategy. This results in the number of nodes in the generated graph not depending on the boundaries, which leads to the generated floorplans deviating from the probability distribution of the original data. Our experiments in the Ablation Studies further corroborated this conclusion.

In our method, we represent room types as node attributes and the connectivity between rooms as edge attributes. CEDM uses node attributes and boundary conditions as guiding conditions, which we represent as $C$. Edge attributes are represented by the spaces $\xi$, with cardinality $b$. As we only consider the presence or absence of edges between nodes, the value of $b$ is set to 2. We use $e_{ij}$ to denote

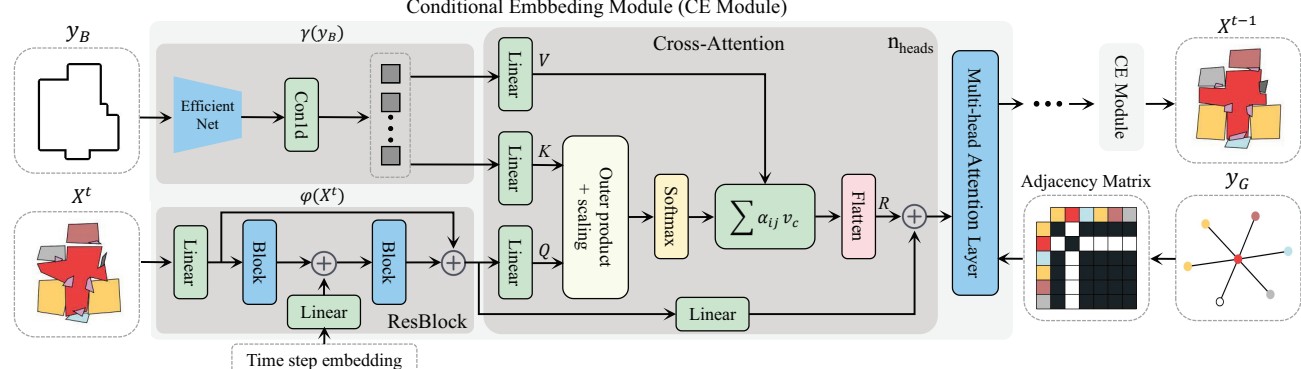

**Figure 4: The overview of CEM-DM. The denoising network takes as input a building boundary $y_B$, a bubble graph $y_G$, and noised floorplan $X^t$. The core module of CEM-DM is a Transformer model with the conditional embedding module. The conditional embedding module employs cross-attention to integrate boundary features extracted by EfficientNet into the vertex features of a noised floorplan. Additionally, the adjacency matrix represented by the generated graph is utilized as a mask matrix in the multi-head attention layers.**

the connection relationship between node $i$ and node $j$, and $E$ to denote all the edge attributes in the graph $G$. Simultaneously, we use $e_{ij} \in \mathbb{R}^b$ to denote the one-hot encoding of edges. A tensor $E \in \mathbb{R}^{n \times n \times b}$ groups the one-hot encoding $e_{ij}$ of each edge, where $n$ denotes the number of nodes.

Similarly to diffusion models for images, which apply noise independently on each pixel, we diffuse separately on each edge attribute. For the noise model, we use the same noise representation as *DiGress*, which is represented by transition matrices $(Q^1, Q^2, ..., Q^T)$ such that $[Q^t]_{ij} = q(e^t = j | e^{t-1} = i)$ represents the probability of edge $e$ transitioning from state $i$ to state $j$ between time $t-1$ and $t$: $q(E^t | E^{t-1}) = E^{t-1}Q^t$. Since the forward process is still Markovian, the transition matrix from $E$ to $E^t$ reads $\bar{Q}^t = Q^1...Q^t$. When $\bar{Q}^t$ is precomputed, the noisy states $E^t$ can be built from $E$: $q(E^t | E) = E\bar{Q}^t$. When conditions $C$ are introduced, the forward diffusion process remains unchanged. This is because the forward diffusion process involves adding noise to the original edge attributions until they become pure noise, the process that is unrelated to the conditions $C$:

$$q(E^t | E^{t-1}, C) = q(E^t | E^{t-1}) = E^{t-1}Q^t. \tag{1}$$

The posterior distribution $q(E^{t-1} | E^t, E, C)$ can also be computed using Bayes rule:

$$q(E^{t-1} | E^t, E, C) \propto E^t(Q^t)' \odot E\bar{Q}^{t-1}, \tag{2}$$

where $\odot$ denotes a elementwise product. Afterward, we can utilize Eq. 2 to calculate the probability distribution of $E^{t-1}$.

For denoising network $\phi_\theta$ parametrized by $\theta$, it takes a noisy edge $E^t$ and conditions $C$ as input and aims to predict the clean edges $p_\theta(E | E^t, C)$. To train $\phi_\theta$ we use the cross-entropy loss $l$ between the predicted edge probabilities $\hat{p}^E$ and the true edge $E$:

$$L(\hat{p}^E, E) = \sum_{1 \leqslant i, j \leqslant n} \text{Cross} - \text{entropy}(e_{ij}, \hat{p}^E_{ij}). \tag{3}$$

Once the diffusion model is trained, we can obtain a pure edge $E$. Then, by using $E$, we can sample to get $E^{t-1}$. To do so, we need

to estimate the reverse diffusion iteration $p_\theta(E^{t-1} | E^t, C)$ using the prediction $\hat{p}^E$. We model this distribution as a product over edges:

$$p_\theta(E^{t-1} | E^t, C) = \prod_{1 \leqslant i, j \leqslant n} p_\theta(e^{t-1}_{ij} | E^t, C). \tag{4}$$

To compute each term, we marginalize over the network predictions:

$$p_\theta(e^{t-1}_{ij} | E^t, C) = \sum_{e \in \xi} p_\theta(e^{t-1}_{ij} | e_{ij} = e, E^t, C)\hat{p}^E_{ij}(e), \tag{5}$$

where we choose:

$$p_\theta(e^{t-1}_{ij} | e_{ij} = e, E^t, C) = q(e^{t-1}_{ij} | e_{ij} = e, e^t_{ij}, C). \tag{6}$$

Eq. 6 is valid only under the condition of $q(e^t_{ij} | e_{ij} = e) > 0$. Finally, the probability distribution of $E^{t-1}$ can be calculated, and thus the $E^{t-1}$ can be sampled, which will be the input of the denoising network at the next time step.

For the CEDM's architecture (see Figure 3), the denoising network takes noisy edge attributions $E^t$, graph-level features $Y$ and conditions $C$ as input and outputs tensors $E$ which represent the predicted distribution over clean edge. We first use *ResNet34* [12] to perform feature extraction on the boundaries and obtain a node sequence by performing one-hot encoding on the node attributes. To improve the network expressivity, we use formulas [28] to calculate the number of cycles in the graph. The obtained feature $y$ and feature $Y$ represent the node-level cycle feature and graph-level cycle feature, respectively. Afterward, we concatenate the boundary features, node sequence, and node-level cycle features as the input conditional features $C$. At the core of the denoising network, we use a modified graph transformer network proposed by [11]. In the transformer block, we first compute the unnormalized attention scores using the updated conditional features $C'$ without applying softmax. Finally, we incorporate both the edge features $E'$ and the graph-level features $Y'$ into the attention scores by employing scale and shift operation [27].

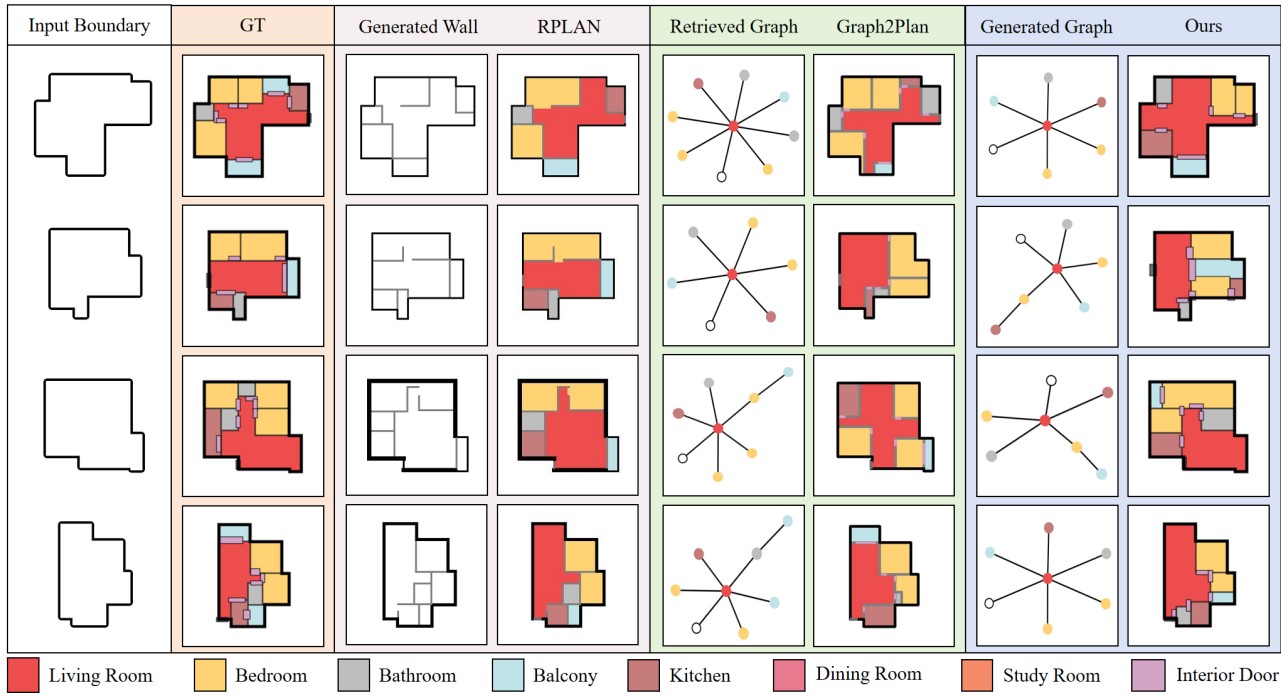

**Figure 5: Comparison to the state-of-the-art with the only boundary. Boundaries are extracted from the training samples to serve as conditional inputs for all methods. Each row displays the results of different methods applied to the same boundary.**

## 3.3 Conditional embedding mechanism

So far, we have proposed a two-stage approach for generating graphs. In the following, we introduce how to use our conditional embedding module to incorporate boundary conditions (denoted as $y_B$) and graph conditions (denoted as $y_G$) into denoising network, named CEM-DM (see Figure 4).

For the boundary constraint, we use a three-channel image as input, which is consistent with the boundary condition representation. First, a modified *EfficientNet-b1* [32] is used to obtain the spatial features. By removing the last fully connected layer and the adaptive average pooling layer, a multi-channel feature map (8x8) is ultimately obtained. It is worth noting that we use the pre-trained *EfficientNet-b1* parameters as the initial parameters for our feature extraction model. Then, we use 1-D convolution to transform feature maps into sequences $\gamma(y_B)$ and employ ResBlock to merge the time step $t$, which has undergone position embedding, with the corner coordinates of $X^t$, thereby obtaining the features $\varphi(X^t)$. The Block in ResBlock utilizes instance normalization (IN) and sigmoid-weighted linear unit (SiLU). Subsequently, we enhance the transformer backbone with the cross-attention layer. To enhance the performance of generating floorplans without boundaries, we add the output of the ResBlock to $R$ in the cross-attention layer. The output result $R$ is ultimately used as the input for the Multi-head Attention Layers.

For the graph constraint, we employ the same method used in *HouseDiffusion*, transforming them into an adjacency matrix and using it as a mask matrix in the Multi-head Attention Layers, limiting attention to connected rooms only. We also use two other

| Condition | Model | FID ($\downarrow$) | BC ($\downarrow$) |
|---|---|---|---|
| | RPLAN | $63.7_{\pm 2.4}$ | $\mathbf{0.0}_{\pm 0.0}$ |
| Boundary | Graph2Plan | $41.3_{\pm 3.0}$ | $0.14_{\pm 0.0}$ |
| | Ours | $\mathbf{8.8}_{\pm 0.3}$ | $0.05_{\pm 0.0}$ |

**Table 1: FID score and BC comparison to *RPLAN* and *Graph2Plan* with the only boundary. 512 generated floorplans are selected to calculate the FID score and BC.**

types of mask matrices: the component-wise mask matrix and the global mask matrix. They are used for limiting attention among corners within the same room and between every pair of corners across all rooms, respectively.

Based on conditioning pairs, we can train the conditional diffusion model via:

$$L = E_{X,\, y,\, \epsilon \sim \mathcal{N}(0,1),\, t} \left[ ||\epsilon - \epsilon_\theta \left( X^t, t, y_B, y_G \right) \,||_2^2 \right]. \quad (7)$$

Because our method considers graphs as necessary conditions and boundaries as optional constraints, we train both with and without boundaries simultaneously in CEM-DM. This is facilitated by randomly assigning a value of 0 to the boundary constraints of some samples in the batch size. During the inference, inspired by *Classifier-free guidance*[14], we introduce a hyperparameter $\lambda \in (0, 1)$ to determine whether to use boundaries. Specifically, Use the following formula to infer:

$$\tilde{\epsilon}_\theta \left( X^t, t, y_B, y_G \right) = \lambda \epsilon_\theta \left( X^t, t, y_B, y_G \right) + (1 - \lambda) \, \epsilon_\theta \left( X^t, t, y_G \right). \quad (8)$$

We also provide a parameter $w \in \{True, False\}$, paired with $\lambda$, to dictate the selection of one of three scenarios for generation.

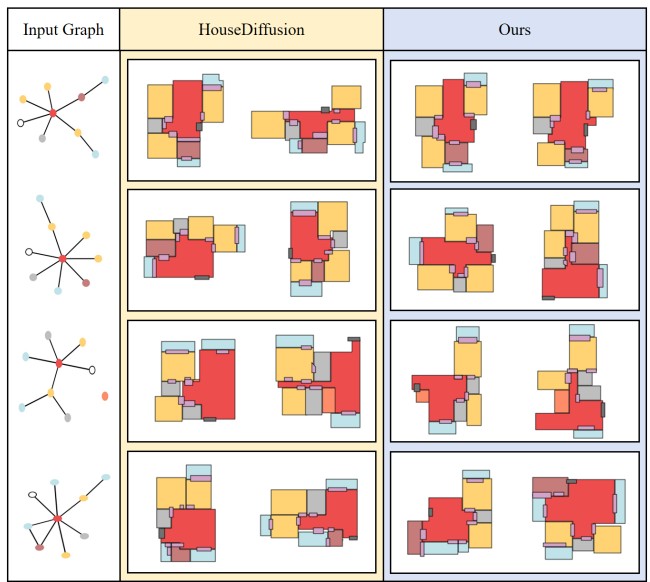

Figure 6: Comparison to HouseDiffusion. An example of graph-constrained floorplan generation using HouseDiffusion and our method is shown. For each graph, each method is executed twice, generating two different floorplans.

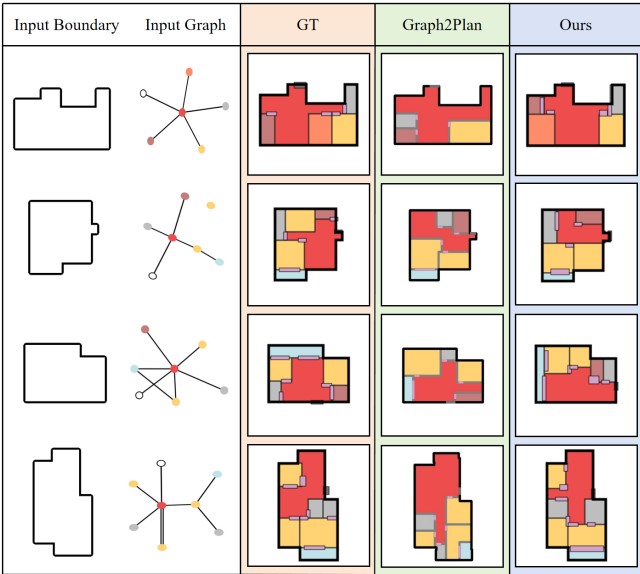

Figure 7: Comparison to Graph2Plan with the same boundary and graph. All conditions are extracted from the ground truth and used as input for both methods.

## 4 EXPERIMENTS

### 4.1 Experimental Setting

**Dataset.** We conduct experiments on a representative benchmark RPLAN [36], which is a large-scale dataset with more than 80K real floorplans from residential buildings. In this dataset, each floorplan is represented as an image, which provides detailed information about the types of rooms and their connectivity relationships.

**Baselines.** We compare our method with the following three state-of-the-art floorplan generation methods: (i) *RPLAN* [36] is a regression model that can generate floorplans through only boundaries; (ii) *HouseDiffusion* [29] only takes graphs as conditions to generate vector floorplans based on diffusion models; (iii) *Graph2Pan* [15] can generate floorplans under both boundaries and graphs.

**Evaluation Metrics.** We conducted both quantitative and qualitative evaluations. For quantitative evaluations, we use two metrics: (i) Diversity – it is evaluated by the Frechet Inception Distance (FID) [13], which is a global metric to calculate the distribution similarity between the generated images and the ground truth; (ii) Compatibility – it includes Boundary Compatibility (BC) and Graph Compatibility (GC). Specifically, BC calculates the area difference between the boundary geometry and the convex hull formed by the geometries of all rooms [31], while GC uses the modified Graph Edit Distance [1] to compare the room connectivity relationships in the graph with those in the generated floorplans.

### 4.2 Implementation Details

We use PyTorch to implement and train all networks. All the experiments are run on a single A800 GPU. Before training. we use post-processing tools to extract boundary images, graphs, and vector floorplans from the RPLAN. The extracted data is used as the dataset for all networks.

In the two-stage approach, 80% of the dataset is used for training and the remaining 20% for testing. For the node prediction networks, our experimental procedure is the same as [36]. For the CEDM, Adam [19] is used as the optimizer with weight decay [23], and we train the network for 1000 epochs with a batch size of 6000. The initial learning rate is set to 2e-4.

For the CEM-DM, similar to the *HouseDiffusion*, the dataset is divided into four groups based on the number of rooms (i.e., 5, 6, 7, or 8 rooms). To generate floorplans for each group, we remove the group's samples from training to prevent the network from just memorizing them. We use the Adam optimizer with default settings combined with an exponential falloff from le-3 to le-5 over 400k steps. We train for 400k steps with a mini-batch of 400 floorplans. We set the number of diffusion steps to 1000 and uniformly sample *t* during training.

### 4.3 Results and Discussions

We compare our method with baselines under three input conditions: only boundaries, only graphs, as well as both boundaries and graphs. Additionally, we also showcase the capability of *Cons2Plan* to generate diverse floorplans.

**Only boundaries.** In this setting, we compare with *RPLAN* and *Graph2Plan* since they can generate floorplans using only boundary conditions. Specifically, *RPLAN* achieves floorplan generation by predicting room positions and generating inner walls, while *Graph2Plan* searches for the closest floorplans to the input boundary in the database and extracts their corresponding graphs to use as its input. In contrast, our method employs a two-stage approach to generate diverse graphs based on the given boundary.

| Condition | Model | FID (↓) | | | | GC (↓) | | | |
|---|---|---|---|---|---|---|---|---|---|
| | | 5 | 6 | 7 | 8 | 5 | 6 | 7 | 8 |
| Graph | HouseDiffusion | $11.2_{\pm 0.2}$ | $\mathbf{10.3}_{\pm 0.2}$ | $\mathbf{10.4}_{\pm 0.4}$ | $\mathbf{9.5}_{\pm 0.1}$ | $1.5_{\pm 0.0}$ | $\mathbf{1.2}_{\pm 0.0}$ | $\mathbf{1.7}_{\pm 0.0}$ | $2.5_{\pm 0.0}$ |
| | Ours w/o rc | $12.9_{\pm 0.1}$ | $13.1_{\pm 0.3}$ | $12.3_{\pm 0.3}$ | $11.9_{\pm 0.4}$ | $1.5_{\pm 0.0}$ | $1.8_{\pm 0.1}$ | $2.2_{\pm 0.1}$ | $2.9_{\pm 0.0}$ |
| | Ours | $\mathbf{9.4}_{\pm 0.2}$ | $10.8_{\pm 0.3}$ | $10.6_{\pm 0.2}$ | $9.9_{\pm 0.1}$ | $\mathbf{1.4}_{\pm 0.1}$ | $1.3_{\pm 0.1}$ | $1.9_{\pm 0.2}$ | $\mathbf{2.2}_{\pm 0.1}$ |

**Table 2: FID score and GC results comparison among HouseDiffusion, our method, and our method without residual connections, with only graphs. 512 generated floorplans are selected to calculate the FID score and GC. For the HouseDiffusion method, we copy the numbers reported in the HouseDiffusion paper [29].**

| Condition | Model | FID (↓) | | | | GC (↓) | | | | BC (↓) | | | |
|---|---|---|---|---|---|---|---|---|---|---|---|---|---|
| | | 5 | 6 | 7 | 8 | 5 | 6 | 7 | 8 | 5 | 6 | 7 | 8 |
| B & G | Graph2Plan | $28.0_{\pm 0.0}$ | $29.5_{\pm 0.0}$ | $31.7_{\pm 0.0}$ | $32.7_{\pm 0.0}$ | $\mathbf{0.5}_{\pm 0.1}$ | $\mathbf{0.7}_{\pm 0.1}$ | $1.6_{\pm 0.3}$ | $3.0_{\pm 0.5}$ | $0.10_{\pm 0.0}$ | $0.11_{\pm 0.0}$ | $0.12_{\pm 0.0}$ | $0.12_{\pm 0.0}$ |
| | Ours | $\mathbf{7.8}_{\pm 0.0}$ | $\mathbf{6.8}_{\pm 0.1}$ | $\mathbf{6.4}_{\pm 0.2}$ | $\mathbf{6.2}_{\pm 0.2}$ | $1.0_{\pm 0.1}$ | $1.2_{\pm 0.1}$ | $\mathbf{1.6}_{\pm 0.0}$ | $\mathbf{2.2}_{\pm 0.1}$ | $\mathbf{0.05}_{\pm 0.0}$ | $\mathbf{0.05}_{\pm 0.0}$ | $\mathbf{0.06}_{\pm 0.0}$ | $\mathbf{0.06}_{\pm 0.0}$ |

**Table 3: FID score, GC, and BC comparison to Graph2Plan with the boundary and graph conditions. 512 generated floorplans are selected to calculate the FID score, GC, and BC.**

Figure 5 and Table 1 show the qualitative and quantitative evaluation, respectively. As shown in Figure 5, *RPLAN* generates crumbling walls in all samples. This is due to the low accuracy of WallNet used in generating semantic images of walls. On the other hand, the biggest issue with *Graph2Plan* is that after selecting the graph for dataset retrieval, some rooms are missing in the generated floorplans (e.g., the balcony is missing in the second and last rows, and the bathroom is missing in the third row). This is because the missing rooms are completely covered by other rooms after post-processing. It is worth noting that our method can not only generate high-quality floorplans but also produce floorplans with different graphs from the ground truth. Table 1 shows an obvious improvement in FID score compared to all other methods. Regarding BC comparison, *RPLAN* is not suitable for comparison since it only generates the center position of rooms and the inner wall position for each room. However, our method still performs better than *Graph2Plan* in terms of BC.

**Only graphs.** We compare the floorplans created using our approach with those produced by *HouseDiffusion* when only graphs are provided. Qualitative and quantitative evaluations are shown in Figure 6 and Table 2, respectively. As shown in Figure 6, the quality of floorplans generated by both *HouseDiffusion* and our method is quite similar. In Table 2, our method performs slightly worse than *HouseDiffusion* in terms of FID score and GC. This is primarily due to CEM-DM training to generate floorplans in a mini-batch that includes both cases with and without boundaries, while *HouseDiffusion* only trains for cases without boundaries. Our approach results in a loss of probability density when predicting only graphs. However, we mitigate this effect by incorporating residual connections. We also present the quantitative results of the method without residual connections, named Ours w/o rc, to demonstrate the effectiveness of our improvements.

**Both boundaries and graphs.** In this setting, we compare with *Graph2Plan* since it can generate floorplans with both boundary and graph conditions. The qualitative evaluation results are shown in Figure 7, from which we can see that *Graph2Plan* still has cases of missing rooms (e.g., the first row is missing a study room). In addition, to test the performance of our method, we selected some

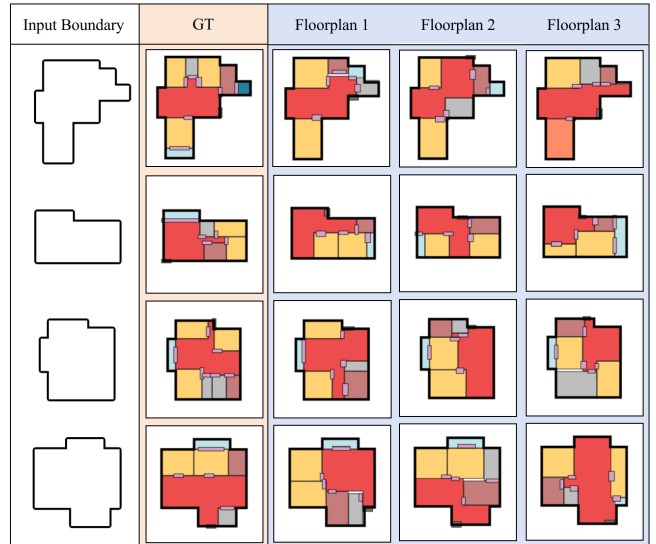

**Figure 8: Performance testing of Cons2Plan in generating floorplans. We executed three independent trials of Cons2Plan with identical boundary conditions, resulting in the generation of three unique floorplans.**

complex graphs and boundaries for the last three rows. In the second row, there is a bedroom that does not connect to any other rooms, and in the last two rows, there are bedrooms connected to two rooms simultaneously. *Graph2Plan* is unable to generate floorplans under such graph conditions. Because it simply places room boxes based on the spatial relationship of nodes in the graph, and the post-processing only adds door decorations. In contrast, our method can generate floorplans that fully satisfy both conditions.

Table 3 shows a quantitative evaluation between the two methods. To ensure fairness, we do not use the post-processing step when calculating the BC for *Graph2Plan*. From Table 3, it can be observed that our method has an obvious improvement in FID score and BC. The FID score of our generated floorplans is reduced by

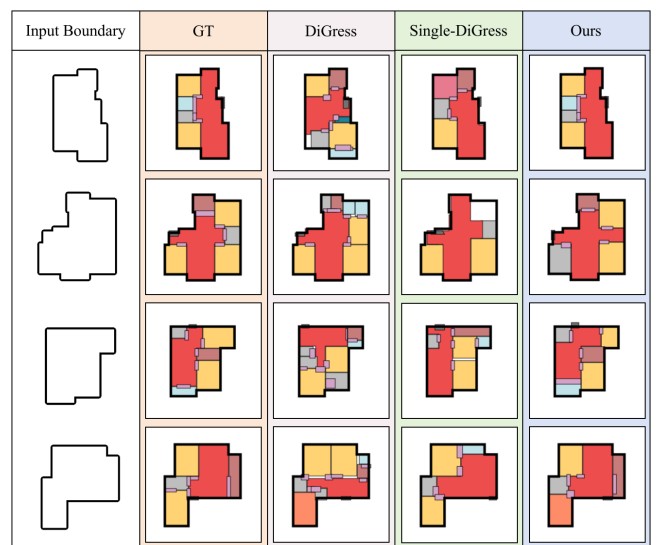

**Figure 9: Ablation studies. We compare our two-stage approach with both the direct application of the DiGress method and the Single-DiGress method. We generate graphs using each method and then use these graphs as conditions to generate floorplans with our CEM-DM.**

an average of 75% compared to *Graph2Plan*. BC is also lower than *Graph2Plan*'s by at least 50% in all cases. Although our method has a higher GC than *Graph2Plan* when the number of rooms is small (e.g., less than 7), as the number of rooms increases, our method demonstrates a clear advantage in GC.

**Floorplan generation performance.** We also test the performance of our *Cons2Plan* to generate diverse floorplans with the only boundaries. The generated results can be seen in Figure 8. We use our approach to generate three different floorplans under the same boundary. In the first two rows, our method can generate floorplans with different room numbers and room connection relationships. In the last two rows, compared to the ground truth, our method can generate floorplans with the same room types but different connection relationships. This demonstrates that our *Cons2Plan* is capable of generating plausible and diverse floorplans through our two-stage approach and the conditional embedding module.

## 4.4 Ablation Studies

We conduct a series of ablation studies to verify the effectiveness of our technical contributions.

**Two-stage approach.** To verify the effectiveness of our two-stage approach, we compared our method with a direct application of *DiGress* to generate graphs. Additionally, we also create a two-stage approach that solely relies on the predicted number of nodes from regression models, while discarding node type information. In generating edges, this approach uses only the boundary image as a constraint, concurrently predicting the probability distributions for both nodes and edges. We named this method *Single-DiGress*.

Figure 9 and Table 4 qualitatively and quantitatively measure the effectiveness of our two-stage approach. Directly using *DiGress* to generate graphs results in the creation of an excessive number

| Condition | Model | FID ($\downarrow$) | BC ($\downarrow$) |
|---|---|---|---|
| | DiGress | $10.3_{\pm0.2}$ | $0.05_{\pm0.0}$ |
| Boundary | Single-DiGress | $9.3_{\pm0.3}$ | $0.05_{\pm0.0}$ |
| | Ours | $\mathbf{8.7}_{\pm0.2}$ | $\mathbf{0.05}_{\pm0.0}$ |

**Table 4: FID score and BC comparison to DiGress and Single-DiGress with the only boundaries. All metrics were calculated based on the 512 floorplans generated.**

| Embedding Method | FID ($\downarrow$) | GC ($\downarrow$) | BC ($\downarrow$) |
|---|---|---|---|
| Simple | $12.0_{\pm0.5}$ | $2.1_{\pm0.1}$ | $0.09_{\pm0.0}$ |
| Ours | $\mathbf{6.8}_{\pm0.1}$ | $\mathbf{1.3}_{\pm0.1}$ | $\mathbf{0.05}_{\pm0.0}$ |

**Table 5: Quantitative evaluation includes the FID score, GC and BC for the two condition embedding methods. Both metrics are calculated using the same 512 ground truth samples.**

of room nodes. This is mainly due to the node acquisition strategy employed by DiGress, leading to poor floorplans that are overly crowded and do not conform to the intuitive principles of actual floorplan designs, as demonstrated in Figure 9. For *Single-DiGress*, while it seemingly is capable of generating reasonable floorplans based on boundary conditions, as shown in Figure 9, it produces a greater variety of node types. This randomness stems from its use of a diffusion model without fully considering the boundaries to generate node types, whereas our method produces node types that are more realistic. Consequently, *Single-DiGress* exhibits a higher FID score than our method. However, both methods show an improvement in FID scores compared to the direct use of the *DiGress* method, as indicated in Table 4.

**Conditional embedding module.** In *Cons2Plan*, we use the conditional embedding module to embed the boundary constraint into the denoising network, guiding the model to generate floorplans that meet the constraints. To demonstrate the effectiveness of this module, we conduct relevant ablation studies and show the results in Table 5. We compare our conditional embedding method with a simple approach of directly adding the boundary features to each corner features $\gamma(y_B) + \varphi(X^t)$. We extract boundaries and graphs from samples as conditional input for evaluation. We can see that simply adding boundary constraints to corner features performs poorly in all evaluation metrics compared to our approach. This confirms the effectiveness of the conditional embedding module in our conditional diffusion model.

## 5 CONCLUSION

This paper proposes *Cons2Plan*, a floorplans generation framework that is more generalizable than SOTA floorplans generation methods in terms of input constraint conditions. The proposed method uses a conditional diffusion model with a conditional embedding module. By combining regression models and a discrete diffusion model, this method not only takes into account the boundary conditions but also significantly improves the diversity of the generated graphs. Extensive experiments demonstrate that our method supports various conditions, producing higher quality floorplans than state-of-the-art techniques.

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
