# OpenReview forum: "Cons2Plan: Vector Floorplan Generation from Various Conditions via a  Learning Framework based on Conditional Diffusion Models"
_acmmm.org/ACMMM/2024/Conference — MM2024 Poster_

### Official Review · Reviewer_tQsk · 2024-05-24

**Rating:** 4
**Confidence:** 3

**Summary:**

The paper introduces Cons2Plan, a novel learning framework for generating high-quality vector floorplans from various input conditions, including graphs, boundaries, or a combination of both. The core component of Cons2Plan is a conditional diffusion model that utilizes a conditional embedding module to incorporate these conditions during the generation process. The framework also includes a two-stage approach for generating graph conditions from boundaries, leveraging three regression models for node prediction and a Conditional Edge Generation Diffusion Model (CEDM) for edge generation. Through extensive experiments, the authors demonstrate that Cons2Plan outperforms state-of-the-art methods like RPLAN and Graph2Plan in both qualitative and quantitative evaluations.

**Strengths:**

1. The introduction of a conditional diffusion model combined with a two-stage approach for handling diverse input conditions is innovative and enhances the versatility of floorplan generation.
2.The generated floorplans are of higher quality compared to those produced by state-of-the-art methods, indicating the robustness and practicality of Cons2Plan.
3.The ability to generate floorplans from multiple types of conditions (graphs, boundaries, or both) adds significant flexibility, making the framework applicable to various real-world scenarios.

**Limitations:**

1.  The use of a carefully designed conditional embedding module and the transformation of the diffusion model into a floorplan generator using cross-attention mechanisms demonstrate a thorough and well-thought-out approach. However, it is easily to consider using a diffusion model in this scenario,the reliance on specific regression models and the CEDM for generating graphs from boundaries may limit the generalizability of the approach to different types of input conditions or datasets. Do you proposed new modules, networks or concepts?

2. The proposed framework is quite complex, involving multiple components and stages, which might make it difficult to implement and replicate without access to specific details and resources. This method is a two-stage framework, could you explain the difficulties or possible solutions to do end-to-end prediction?


3. While the method shows promise, its scalability to very large or very complex floorplan generation tasks has not been fully addressed .In the abstract, authors stated "However, generating floorplans that satisfy
various conditions remains a challenging task", could you explain more about the various conditions, and why it would be challenges?

4. Although the paper compares Cons2Plan with some state-of-the-art methods, additional comparisons with a broader range of existing techniques could further strengthen the evaluation. The comparison baseline methods is limited in Table 1.

5. Figure 6 does not show the ground truth, it is hard to distinguish which results is better? maybe the ground truth or metrics can be attached.

Anyway, this paper aims to tackle a very interesting problem (real scenario), thus, I hold a positive attitude.

**Suitability:**

2

---

### Official Review · Reviewer_CDsr · 2024-05-25

**Rating:** 4
**Confidence:** 3

**Summary:**

This paper propose Cons2Plan, a novel learning framework for vector floorplan generation through the conditional diffusion models that generates vector floorplans under three different input conditions. To support this, the authors constructed a two-stage approach for graph conditions generation by incorporating three regression models for node prediction and a novel conditional edge generation diffusion model. The experiments demonstrated Cons2Plan’s superior performance compared to existing frameworks

**Strengths:**

- This paper propose a novel framework for vector floorplan generation that combines condition diffusion model as a floorplan generator and regression models, providing a more generalizable floorplans generation methods.

- Organized results and discussions. Cons2Plan presented performance improvements for various conditions.

**Limitations:**

- In Table 2, L739 “This is primarily due to CEM-DM training to generate floorplans in a mini-batch that includes both cases with and without boundaries, while HouseDiffusion only trains for cases without boundaries.” It would be more convincing if the result of Cons2Plan only trains for cases without boundaries is provided.

- The visualization comparison figures are hard to distinguish their performance directly. The qualitative evaluation conclusion mentioned such as L752 “the first row is missing a study room” are suggested to be marked in the images.

**Suitability:**

3

---

### Official Review · Reviewer_159L · 2024-05-31

**Rating:** 4
**Confidence:** 3

**Summary:**

This paper introduces Cons2Plan, a learning framework designed to automatically generate high-quality vector floorplans from a variety of input conditions, including graphs, boundaries, or a combination of both. The framework features a two-stage approach for generating graph conditions from boundary inputs, involving regression models for node prediction and a conditional edge generation diffusion model (CEDM) for edge generation. The paper presents extensive evaluations, demonstrating that Cons2Plan outperforms existing state-of-the-art methods in generating diverse and high-quality floorplans under various conditions.

**Strengths:**

1. The paper presents an innovative approach for conditional floorplan generation by integrating a conditional diffusion model with a conditional embedding module, representing a significant advancement in the field of automated floorplan generation.

2. The paper provides a meticulous evaluation of the Cons2Plan framework, offering detailed qualitative and quantitative analyses in comparison to leading methods, as well as insightful ablation studies. These evaluations effectively validate the framework's ability to generate higher quality floorplans than existing state-of-the-art methods.

3. The incorporation of boundary information as a condition has proven to be a meaningful choice, as demonstrated by notable enhancements in the quality of the generated floorplans in the examples provided in the paper.

**Limitations:**

1. The improvements in the graph conditional image generation task appear to be marginal, with the model performing even worse than HouseDiffusion in terms of FID score. Although Cons2Plan leverages an additional module for graph generation compared to HouseDiffusion, the benefits of this module are not clearly articulated.

2. The two-stage graph generation approach in Cons2Plan, relying on a graph generation diffusion model, may result in increased time consumption during inference. The computational intensity of diffusion models raises concerns about the efficiency of the framework during inference, which has not been adequately addressed. An analysis of the trade-offs between the quality of the generated floorplans and the time required to generate them would be beneficial for practical applications.

3. The paper emphasizes the importance of the conditional diffusion model with an enhanced conditional embedding module as the core of the Cons2Plan framework. However, the description of the architecture of CEM-DM is limited, leaving uncertainty about whether the diffusion model in Cons2Plan directly adopts the continuous and discrete diffusion model proposed by HouseDiffusion or introduces new modifications. A more comprehensive explanation of CEM-DM, including any unique features or improvements over HouseDiffusion's approach, would provide a clearer understanding of the framework's contributions.

**Suitability:**

2

---

### Official Review · Reviewer_kv5Z · 2024-05-31

**Rating:** 3
**Confidence:** 3

**Summary:**

This paper proposes a new model for floorplan generation, called Cons2Plan, which uses graph and boundary as conditional information for diffusion model. Specifically, Cons2Plan builds on the existing model HousDiffusion, adding the use of boundaries as conditions to guide the generation of diffusion models. This paper is based on a two-stage approach that first utilizes three regression models for node prediction and a new conditional edge generation diffusion model called CEDM for edge generation. By comparing qualitatively and quantitatively on a benchmark dataset with two existing baselines, the proposed model can improve the quality and diversity of the generated graphs.

**Strengths:**

1. This paper is an interesting work, which proposes the use of Graph and Boundary together as conditional information for diffusion model to generate floorplan.
2. This paper shows technical and theoretical soundness. This paper contains sufficient experimental results, and the proposed model shows improvement compared with the baseline including the graph-conditional models and the a combination of both models.
3. The proposed model of using multiple data sources as diffusion model's conditional information is inspiring.

**Limitations:**

1. The overall presentation of this paper needs improvement, particularly in language expression, logic, and visualization. For instance, the problem definition is unclear, there are unnecessary definitions, and the motivation is not well explained. The writing of the paper requires enhancements.
2. The proposed model lacks a detailed explanation. The authors should provide more information about the model, including its architecture, the training or inference process, and the rationale for using this design.
3. This paper only compares with two baseline models, which is insufficient and unconvincing. The authors should compare with more models to demonstrate the effectiveness of the proposed model.
4. This paper appears to offer limited improvements on House Diffusion, and does not achieve superior results despite using more realistic conditions.

**Suitability:**

2

---

### Meta-Review · Area_Chair_L46p · 2024-07-05

**Recommendation:** Accept (Poster)
**Confidence:** 4

**Metareview:**

This paper addresses an interesting and practical problem: floorplan generation from various input conditions, including graphs, boundaries, or a combination of both. All reviewers agree that the proposed approach is novel and inspiring, and the empirical evaluation is generally comprehensive and convincing. The inclusion of boundary conditions during generation is particularly effective. Like most reviewers, the AC leans towards accepting this paper.

However, as noted by the reviewers, the paper could be improved by providing a clearer, more accessible description of the model design, and a discussion and analysis of the incurred overhead in time complexity. Additionally, better highlighting the visualization results would help demonstrate the advantages of the proposed method. It is strongly recommended that the authors incorporate the reviewers' comments and rebuttal discussion into the final revision of the paper.